# Can use of the serum anti-PLA$_2$R antibody negate the need for a renal biopsy in primary membranous nephropathy?

Omar Ragy[1,2]☯*, Vilma Rautemaa🄌[1]☯, Alison Smith[3], Paul Brenchley[2,4], Durga Kanigicherla[1,2], Patrick Hamilton[1,2,4]

1 Manchester Institute of Nephrology and Transplantation, Manchester University NHS Foundation Trust, Manchester, United Kingdom, 2 Wellcome Centre for Cell-Matrix Research, Division of Cell Matrix Biology and Regenerative Medicine, School of Biological Sciences, Faculty of Biology, Medicine and Health, The University of Manchester, Manchester, United Kingdom, 3 Academic Unit of Health Economics, Leeds Institute of Health Sciences, University of Leeds, Leeds, United Kingdom, 4 Manchester Academic Health Science Centre (MAHSC), The University of Manchester, Manchester, United Kingdom

☯ These authors contributed equally to this work and considered as joint primary authorship.
* omar.ragy@mft.nhs.uk

**Data Availability Statement:** Data restrictions has been imposed by Manchester Foundation Trust Research and Innovation department. Data can be requested through our research and innovation

## Abstract

### Background

Since the emergence of the anti-PLA$_2$R antibody (PLA$_2$R-Ab) test, nephrology practice has not changed dramatically, with most nephrologists still relying on a kidney biopsy to diagnose membranous nephropathy. In this study, we examined the clinical accuracy of the anti-PLA$_2$R antibody test using ELISA in routine clinical care.

### Methods

We conducted a retrospective analysis of PLA$_2$R-Ab testing in 187 consecutive patients seen at a single UK centre between 2003 and 2020. We compared the kidney biopsy findings with the PLA$_2$R-ab antibody test. Patients' demography, urine protein creatinine ratios, serum albumin, and treatment characteristics including supportive and immunosuppressive treatment were recorded. The clinical accuracy of the test (e.g. sensitivity and specificity, positive [PPV] and negative [NPV] predictive values) was calculated using the kidney biopsy findings as the diagnostic reference.

### Results

Mean levels of PLA$_2$R-Ab titre in primary membranous nephropathy were 217RU/ml in comparison to 3RU/ml for both secondary membranous nephropathy and other diagnoses. Most patients with a positive PLA$_2$R-Ab test had a confirmed renal biopsy diagnosis of primary membranous nephropathy with: PPV of 97.3%, sensitivity 75.5%, NPV was 79.8% and specificity was 97.8% at a cut-off threshold of >20 RU/ml.

department in Manchester Foundation trust (R&D. Applications@mft.nhs.uk).

**Funding:** The authors received no specific funding for this work.

**Competing interests:** The authors have declared that no competing interests exist.

## Conclusion

The anti-PLA$_2$R antibody test is a highly specific test for diagnosing membranous nephropathy, and the test has the potential to allow for the diagnosis and treatment in up to 75% of PMN cases without the need for a renal biopsy. Nevertheless, patients with negative PLA2R-Ab tests will still require a biopsy to confirm their diagnosis.

## 1. Introduction

Membranous nephropathy (MN) is one of the leading causes of nephrotic syndrome in adults worldwide [1]. Some cases can be secondary to malignancy, systemic autoimmune disease, drugs, or infections, but in the majority (~80%), it is a primary autoimmune disease caused by antibodies targeting the podocytes. The most common target autoantigen is the M-type phospholipase A2 receptor 1 (PLA$_2$R), identified by Beck *et al* in a pivotal study in 2009, found in up to 70% of patients with primary MN (PMN) [2]. Serum anti-PLA$_2$R autoantibody (PLA$_2$R-Ab) titres have shown promise as a highly specific diagnostic and prognostic biomarker [3, 4], and circumstantial evidence suggests the antibody is pathogenic. High titres are associated with active disease, a higher probability of disease relapse and a lower risk of remission following treatment. The converse is equally true with low titres associated with a higher probability of remission meaning its use as a biomarker for disease monitoring is particularly valuable [5–7].

Currently, clinical practice hinges on a renal biopsy to aid or confirm a diagnosis of primary MN (PMN). The wide availability and high specificity and sensitivity of the serum PLA$_2$R-Ab test has led many clinicians to start considering the use of the PLA$_2$R-Ab test as a diagnostic tool, particularly with the available evidence suggesting that amongst patients with positive PLA$_2$R-Ab test and preserved renal function, a renal biopsy may not provide significant additional information that would alter management. However, evidence is limited and currently based on small numbers [8], with further evidence required to convince nephrologist to alter their routine clinical practice.

## 2. Methods

### 2.1 Data collection

All patients, over the age of 18, with a renal biopsy and a PLA$_2$R-Ab test requested at the Manchester Royal Infirmary between January 2003 and January 2020 were eligible for the study. Samples were excluded if there was no reported PLA$_2$R-Ab test result, or if the antibody test was collected more than 6 months post-renal biopsy. If there were multiple samples from the same patient, the antibody result closest to the biopsy date was chosen and the other results were excluded. Patient demographics, renal biopsy findings, urine protein: creatinine ratio (UPCR) and serum albumin at the time of the PLA$_2$R-Ab test, use of immunosuppressants and renin-angiotensin system inhibitors (RAASi) were also recorded. Medical notes were reviewed by two independent clinicians to identify secondary causes of MN.

From 2017 onwards, all PLA2R-Ab tests were performed by the Sheffield Protein Reference Unit using the Euroimmun ELISA kit with levels <14 RU/ml interpreted as negative, 14–20 RU/ml as borderline, and >20 RU/ml as positive. For samples taken prior to 2017, all anti-PLA$_2$R antibody ELISA tests were performed using the in-house Manchester ELISA as described previously [7].

## 2.2 Ethics

After discussion with the Manchester University Hospitals NHS Foundation Trust Research and Innovation department), the study was exempt from requiring a specific ethical approval as it was considered a retrospective audit as per the Health Research Authority definition. The ethics committee therefore waived the requirement for informed consent and no consent or regulatory research approvals were required or obtained prior to this project commencing.

## 2.3 Statistical analysis

Statistical analysis was performed using GraphPad Prism version 8 (GraphPad Software Inc, San Diego, CA). Statistical significance was set at 0.05.

Normality checked using shapiro-wilk test, and as all parameters significantly deviated from a normal distribution, continuous data are presented as median (interquartile range) and categorical data presented as number and percentage.

The receiver operating characteristic (ROC) curve was generated using established methods and the diagnostic sensitivity and specificity of the anti-PLA2R antibody test were calculated, using renal biopsy as the diagnostic reference standard [9–13]. 95% confidence intervals (CI) calculated using Wilson procedure with correction for continuity (Website http://vassarstats. net/prop1.html) [14, 15].

For analysis, all antibody titres reported as <3 or >3000 RU/ml were converted to 3 and 3000 RU/ml, respectively.

## 3. Results

### 3.1 Patient characteristics

In total, 187 adult patients who had an anti-PLA$_2$R antibody test before, or within 6 months after a renal biopsy, were included in the study (Fig 1). This included 94 patients with PMN, 7 with secondary MN (SMN), and 86 with other diagnoses. The most common other diagnoses for patients who had an PLA$_2$R-Ab test were minimal change disease, focal segmental glomerulosclerosis, IgA nephropathy, diabetes, and hypertension (Table 1). A significantly higher proportion of patients with PMN were male compared to patients with other diagnoses (72% vs 57% respectively, p<0.05).

### 3.2 PLA$_2$R antibody titres

The median antibody titre for patients with PMN was 219 (3–3000), with SMN 3 (3–12), and for other diagnoses 3 (2–36) RU/ml (Fig 2). Of 94 patients with PMN, 71 (76%) had a positive result. One patient with diffuse proliferative glomerulonephritis had an antibody titre of 31 RU/ml, and one patient with focal segmental glomerulosclerosis had a titre of 36 RU/ml. Apart from these two patients, nobody with a diagnosis other than PMN had a positive antibody test result.

### 3.3 Diagnostic sensitivity and specificity

The specificity of the anti-PLA$_2$R ELISA was 0.978 with a sensitivity of 0.755, whilst the Positive predictive value (PPV) was 97.3% and the negative predictive value (NPV) was 79.8% (Table 2). The PLA$_2$R-AB test has high specificity at the current threshold of 20 RU/ml (98%), however, the sensitivity is lower at 75%. Youden's J statistic was highest at titre >10 RU/ml, but at levels >40 RU/ml specificity was 1 (Fig 3 and Table 3).

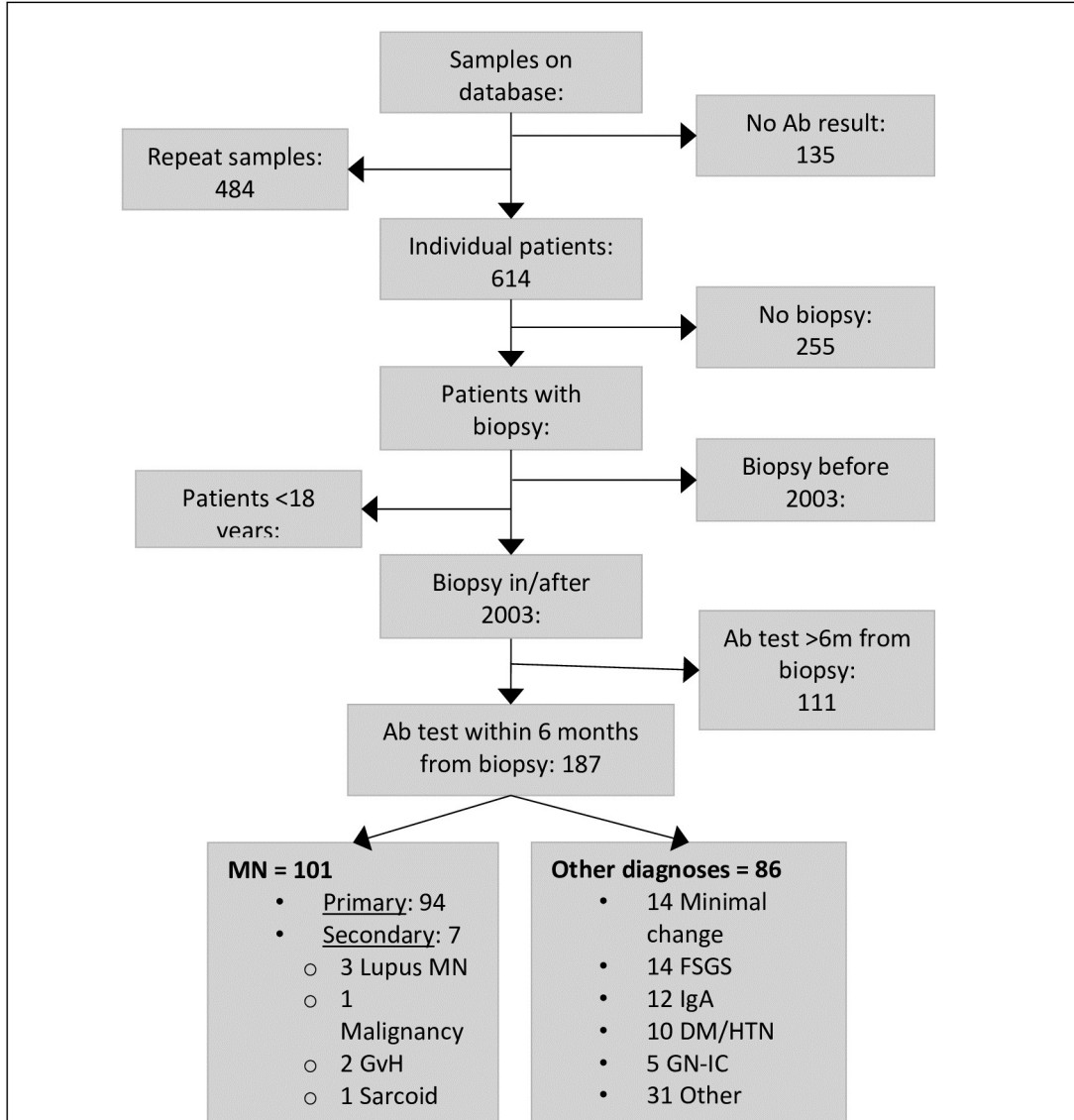

**Fig 1. This is the workflow displaying inclusion and exclusion criteria.** GvH: Graft versus Host, FSGS: focal segmental glomerulosclerosis, IgA: IgA nephropathy, DM: diabetes mellitus, HTN: hypertension, GN-IC: immune complex glomerulonephritis.

### 3.4 Renal function and medication

At the time of the PLA$_2$R-Ab test, patients with PMN had significantly higher uPCR and lower serum albumin than patients with other diagnoses (p<0.001 for both) (Table 1). During their treatment, 90% of patients with PMN received RAASi, compared to 53% of patients with other diagnoses (p<0.001). The use of immunosuppressants was also significantly higher in the PMN group, at 78%, compared to 36% of patients with other diagnoses (p<0.001). Patients with PMN who received immunosuppressants had significantly higher antibody titres than patients who did not receive immunosuppressant treatment (median 361 vs 42 RU/ml respectively, p<0.001). There was no significant correlation between antibody titre and renal function, UPCR and serum albumin at the time of biopsy in the PMN group.

Table 1. These are the patients' demographics. Continuous data are presented as median (range). UPCR: urine protein creatinine ratio, RAS: Renin Angiotensin System.

| | Total | PMN | SMN | Other |
|---|---|---|---|---|
| Patients (n) | 187 | 94 | 7 | 86 |
| Age at PLA2R test | 56 (18–88) | 58 (23–86) | 50 (18–75) | 54 (18–88) |
| Gender (male) | 122 (65%) | 68 (72%) | 5 (71%) | 49 (57%) |
| Ethnicity (white) | 143 (76%) | 77 (82%) | 5 (71%) | 61 (71%) |
| PLA2R Ab Positive | 73 (39%) | 71 (76%) | 0 (0%) | 2 (2%) |
| UPCR at PLA2R test (mg/mmol) | 603 (8–3976) | 768 (8–2390) | 435 (92–3976) | 481 (31–2514) |
| Serum albumin at PLA2R test (g/L) | 21 (6–43) | 19 (6–42) | 18 (7–34) | 27 (7–43) |
| RAS inhibition | 137 (73%) | 85 (90%) | 6 (85%) | 46 (53%) |
| Immunosuppression | 109 (58%) | 73 (78%) | 5 (71%) | 31 (36%) |

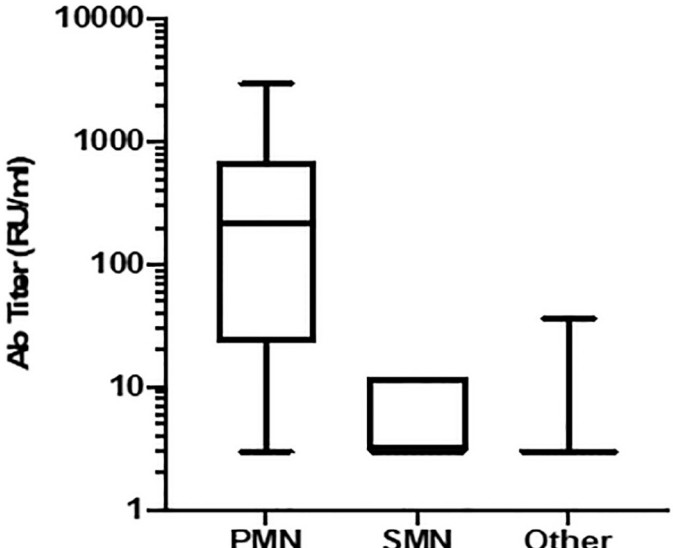

Fig 2. These are antibody titers (RU/ml) with IQR and range, shown on Log$_{10}$ scale.

Table 2. This is the 2*2 diagnostic accuracy table. Sensitivity, specificity, positive predictive value (PPV), and negative predictive value (NPV) of the anti-PLA$_2$R ELISA with antibody titer ≥20 RU/ml interpreted as positive.

| | | Confirmed diagnosis | | |
|---|---|---|---|---|
| | | Primary (PMN) | Not PMN (i.e. SMN or other) | |
| Anti-PLA2R Test result | Positive | 71 | 2 | PPV |
| | | | | 71/73 = **0.973** |
| | | | | (95% CI 0.896–0.995) |
| | Negative | 23 | 91 | NPV |
| | | | | 91/114 = **0.798** |
| | | | | (95% CI 0.711–0.865) |
| | | Sensitivity | Specificity | |
| | | 71/94 = **0.755** | 91/93 = **0.979** | |
| | | (95% CI 0.654–0.836) | (95% CI 0.917–0.996) | |

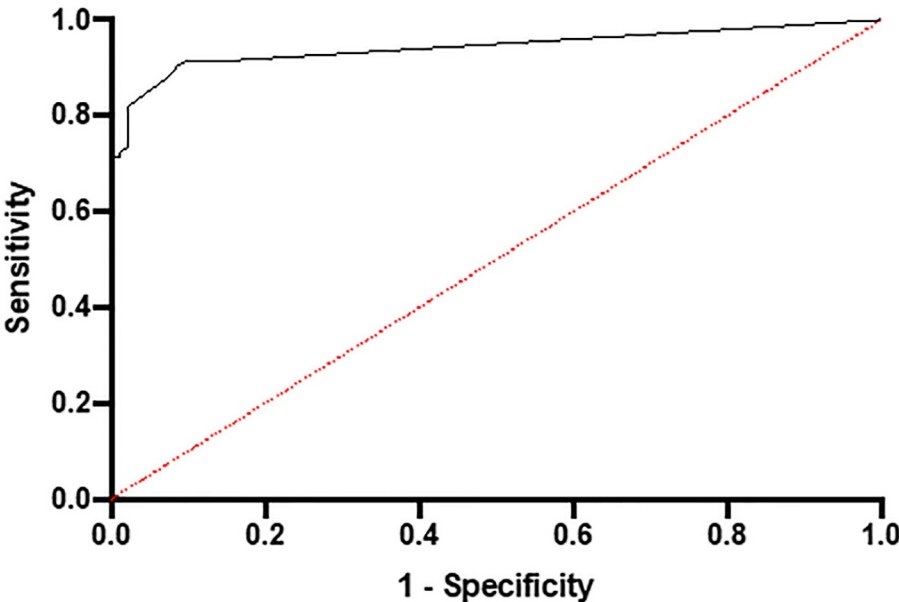

**Fig 3. This is a ROC curve.** It shows the sensitivity and specificity of the serum PLA2R antibody test in our study. Area under curve: 0.943, p<0.0001.

## 4. Discussion

Current management of patients with suspected MN has changed little over the last 20 years. The vast majority of patients undergo a diagnostic renal biopsy with or without further screening investigations. With the discovery of the anti-PLA$_2$R antibody and the increasing understanding of its role in the disease pathogenesis, a more pragmatic patient-focused pathway becomes eminently possible. Our results show that the PLA$_2$R-Ab test had a specificity of 97.8% and sensitivity 76% for biopsy-confirmed PMN, with a titre of >20 RU/ml considered positive. This is in line with a previous meta-analysis from 2014 that found 99% specificity and 78% sensitivity for PMN [16]. The high specificity of the test suggests that we can rely solely on a positive test to diagnose and treat PMN without the need to perform a renal biopsy [17].

Despite its relative safety, an invasive procedure such as a renal biopsy comes with inherent risk of complications, in particular bleeding. The risk of bleeding is raised in those undergoing an emergency biopsy compared to an elective one [18]. Another important consideration is the cost of performing a biopsy. In England the national average unit cost of performing a renal biopsy in the NHS is £774, compared to a cost of £27 for an ELISA PLA$_2$R-Ab test [19]. This does not include any downstream costs brought on by complications, or societal costs such as taking time off work for the procedure, or those incurred by carers. The ability to safely

**Table 3. Sensitivity, specificity and Youden's index of the anti-PLA2R test in diagnosing PMN in this study, over varying cut-off thresholds.**

| Ab Titer (RU/ml) | Sensitivity | Specificity | Youden's J |
|---|---|---|---|
| >10 | 0.883 | 0.924 | 0.808 |
| >20 | 0.756 | 0.978 | 0.734 |
| >30 | 0.734 | 0.978 | 0.713 |
| >40 | 0.691 | 1 | 0.691 |

avoid a biopsy in even a proportion of patients with MN therefore has implications for not only a patient's quality of life but also healthcare systems in general.

One of the main disadvantages of PLA$_2$R-Ab is its low sensitivity. In our study 23 (24%) patients with biopsy-confirmed PMN had a negative PLA$_2$R-Ab test. This means that although the PLA$_2$R-Ab test is very good at ruling in PMN, a negative result cannot be used to rule out PMN as a diagnosis, and renal biopsy is indicated in these patients. There are several reasons why a patient with PMN can test negative for PLA$_2$R-Ab. Firstly, the patient may be in immunological remission, whether spontaneously or due to treatment [20]. Secondly, given the high affinity the anti-PLA$_2$R antibody has for the antigen, seronegativity in clinically active disease states can be attributed to an immune sink phenomenon [21]. Here, the antigen needs to be saturated with immune-complex deposition before there is a detectable circulating level [22]. In these situations, there is some evidence to show staining for the PLA$_2$R antigen on kidney biopsy may be of benefit in seronegative patients [23]. Thrombospondin 7A antigen (THSD7A) was found to be the target in 1–5% of PMN. Even more recently, Exostosin 1 and 2 were found to be target antigens in primary membranous nephropathy [24]. This makes the diagnosis of membranous nephropathy with a negative PLA$_2$R-Ab test still potentially valid, and in these situations, a biopsy warranted.

At present there are two generally available methods of assessing serum PLA$_2$R-Ab: immunofluorescence test (IFT) and the Euroimmun ELISA test [25]. Studies have shown a high concordance between these two tests with high specificity for PMN, especially when used together [8, 26, 27]. The ELISA test provides a quantitative result that can be used to guide treatment, whereas IFT is only semiquantitative but more sensitive at low antibody titres [20]. ELISA and IFT can be used together to increase diagnostic accuracy at low antibody titres and could be useful for patients where there is a high clinical suspicion of PMN but low or negative PLA$_2$R-Ab test.

Studies have shown that some patients with SMN can also have positive PLA$_2$R-Ab, especially SMN associated with hepatitis B infection [28]. This should be taken into account when diagnosing patients using the PLA$_2$R-Ab test, and screening for secondary causes of MN should be carried out as clinically indicated. However, in this study all patients with SMN had a negative PLA$_2$R-Ab result, indicating it is highly specific for PMN.

Anti-PLA$_2$R antibody levels have been shown to correlate with disease severity and clinical response. Patients with very high antibody titres have been found to have more significant reduction in renal function and lower remission rate than patients with lower titres [6, 29].

This study has a number of limitations. This was a retrospective cohort study, so data collection was limited to what was available on electronic patient records. Variability in the timing of biopsies and antibody testing, and other blood tests, has made the interpretation of the results more difficult. Many patients included in the study had their biopsy and PLA$_2$R-Ab test on different days, sometimes months apart. We have included renal function tests done on the day of the biopsy, which makes it difficult to correlate PLA$_2$R-Ab levels when performed at a different timing. A prospective study would be ideal to combat these limitations, however not practical due to the low incidence rate of the disease.

## 5. Conclusion

We have shown that a positive serum anti-PLA$_2$R antibody test has high specificity for PMN. This has the potential to allow for the diagnosis and treatment in up to 75% of PMN cases without the need for a renal biopsy. Patients with negative PLA$_2$R-Ab tests will still require a biopsy to confirm their diagnosis, and further screening for secondary causes of MN should also be considered in specific cases. But in a subset of patients, the ability to avoid an invasive

procedure can provide a benefit to the patients journey and allow for significant cost savings to healthcare providers.

## Author Contributions

**Conceptualization:** Omar Ragy, Patrick Hamilton.

**Data curation:** Vilma Rautemaa, Durga Kanigicherla.

**Formal analysis:** Omar Ragy, Vilma Rautemaa, Alison Smith, Patrick Hamilton.

**Investigation:** Alison Smith.

**Methodology:** Alison Smith, Patrick Hamilton.

**Project administration:** Omar Ragy.

**Supervision:** Paul Brenchley, Durga Kanigicherla, Patrick Hamilton.

**Validation:** Alison Smith.

**Visualization:** Alison Smith, Durga Kanigicherla.

**Writing – original draft:** Omar Ragy, Vilma Rautemaa.

**Writing – review & editing:** Omar Ragy, Vilma Rautemaa, Alison Smith, Durga Kanigicherla, Patrick Hamilton.

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
