## [Decision Letter · Decision Letter 0]

14 Feb 2022

PONE-D-21-33835Can Use of the serum anti-PLA2R antibody negate the need for a renal biopsy in Primary Membranous Nephropathy?PLOS ONE

Dear Dr. Ragy,

Thank you for submitting your manuscript to PLOS ONE. After careful consideration, we feel that it has merit but does not fully meet PLOS ONE’s publication criteria as it currently stands. Therefore, we invite you to submit a revised version of the manuscript that addresses the points raised during the review process.

Please, espond accurately to all the issues of the reviewers. Since the study is limited by the variability in the timing of biopsies and of antibody testing and by nuance in individual biopsies of MN that may impact accuracy of serum based testing, the conclusion of the paper should be less categorical but focusing to the importance of the results in the proceedings of MN diagnosis. 

We look forward to receiving your revised manuscript.

Kind regards,

Fabio Sallustio, PhD

Academic Editor

PLOS ONE

Journal Requirements:

2. In ethics statement in the manuscript and in the online submission form, please provide additional information about the patient records/samples used in your retrospective study. Specifically, please ensure that you have discussed whether all data/samples were fully anonymized before you accessed them and/or whether the IRB or ethics committee waived the requirement for informed consent. If patients provided informed written consent to have data/samples from their medical records used in research, please include this information.

No funding 

No Authors have competing interests

5. Please note that in order to use the direct billing option the corresponding author must be affiliated with the chosen institute. Please either amend your manuscript to change the affiliation or corresponding author, or email us at plosone@plos.org with a request to remove this option.

Additional Editor Comments:

Since the study is limited by the variability in the timing of biopsies and of antibody testing and by nuance in individual biopsies of MN that may impact accuracy of serum based testing, the conclusion of the paper should be less categorical but focusing to the importance of the results in the proceedings of MN diagnosis.

Reviewers' comments:

Reviewer's Responses to Questions

**Comments to the Author**

1. Is the manuscript technically sound, and do the data support the conclusions?

Reviewer #1: Yes

Reviewer #2: Partly

2. Has the statistical analysis been performed appropriately and rigorously? 

Reviewer #1: Yes

Reviewer #2: Yes

3. Have the authors made all data underlying the findings in their manuscript fully available?

Reviewer #1: Yes

Reviewer #2: Yes

4. Is the manuscript presented in an intelligible fashion and written in standard English?

Reviewer #1: Yes

Reviewer #2: Yes

5. Review Comments to the Author

Reviewer #1: Interesting paper

Some issues.

Methods: due to reduced sample size, it should be added if normal distribution was checked for

Methods: was sample sixe calculation performed

Methods; was diagnosis at biopsy confitmed among different operators?

methods: do authors think thanks these results may be of clinical utility?

discussion; do authors think they enrolled enough patients?

Reviewer #2: Thank you for this opportunity to review your original research. I wholeheartedly agree with the authors that medical renal biopsy is an invasive procedure and carries inherent risk of adverse outcome. Thus, non-invasive study is ideal for patients who are poor candidates of biopsy.

In your study of serum PLA2R-Ab testing as potential replacement of biopsy, these are points that need clarification or would benefit from consideration:

1) Please elaborate on how primary MN (PMN) is defined. Patient's records were reviewed by nephrologists but did each patient have complete workup to exclude potential causes of secondary MN? Were histopathologic features of kidney biopsy considered (e.g. immunofluorescence staining pattern, location(s) of immune deposits, presence of other concomitant renal lesions)?

2) Any data on concordance rate of serum PLA2R-ab vs. tissue PLA2R staining on biopsy specimen? Any cases exhibiting "immune sink" phenomenon?

3) Renal biopsies are obtained for both diagnostic and prognostic purposes. Thus, it would be worthwhile to investigate the performance of serum PLA2R-Ab while considering other parameters (e.g. %global and segmental glomerulosclerosis, % tubulointerstitial scarring, tubulointerstitial inflammation, Ehrenreich-Churg stage of MN)

6. PLOS authors have the option to publish the peer review history of their article (what does this mean?). If published, this will include your full peer review and any attached files.

Reviewer #1: **Yes: **Fabrizio D'Ascenzo

Reviewer #2: No

---

## [Author Response · Author response to Decision Letter 0]

27 Apr 2022

First reviewer’s questions:

1) Methods: due to reduced sample size, it should be added if normal distribution was checked for

Added this sentence in the statistical analysis section:

Normality checked using shapiro-wilk test, and as all parameters significantly deviated from a normal distribution, continuous data are presented as median (interquartile range) and categorical data presented as number and percentage. 

2) Methods: was sample size calculation performed?

As this was a retrospective single centre analysis, the sample size was determined by the data we had available to us. We have not conducted any hypothesis tests, but we have added in 95% confidence interval estimates to clearly convey the uncertainty around our estimates.

The manuscript has been updated to reflect this. The receiver operating characteristic (ROC) curve was generated using established methods and the diagnostic sensitivity and specificity of the anti-PLA2R antibody test were calculated, using renal biopsy as the diagnostic reference standard (9,10,11,12,13). 95% confidence intervals (CI) calculated using Wilson procedure without correction for continuity 

Ref – Website http://vassarstats.net/prop1.html accessed on 29/03/22

And the references from the website added:

Newcombe, Robert G. "Two-Sided Confidence Intervals for the Single Proportion: Comparison of Seven Methods," Statistics in Medicine, 17, 857-872 (1998).

Wilson, E. B. "Probable Inference, the Law of Succession, and Statistical Inference," Journal of the American Statistical Association, 22, 209-212 (1927).

3) Methods: was diagnosis at biopsy confirmed among different operators?

 All biopsies were reported by a Renal Histopathologist and discussed at the departmental biopsy review MDT.

4) Methods: do authors think thanks these results may be of clinical utility?

 We believe that those results will be a steppingstone to move away from performing renal biopsies for patients with a positive PLA2R ab test 

5) Discussion: do authors think they enrolled enough patients?

As far as we are aware, this is the largest retrospective study to address that question. Given that MN is listed on the National Registry of Rare Kidney Disease (RaDaR), so we think that the number of patients enrolled to answer our question was enough 

Second reviewer’s questions:

1)Please elaborate on how primary MN (PMN) is defined?

PMN was confirmed based on a renal biopsy, having a positive anti PLA2R ab test and negative secondary work up.

2)Patient's records were reviewed by nephrologists but did each patient have complete workup to exclude potential causes of secondary MN? 

Secondary workup for cancers was only performed for certain high-risk categories (older than 50 years old, smokers, history of change in bowel habits and signs of iron deficiency anaemia), atypical findings on histology or based on full history and examination. PSA for those >50 years old and CT TAP/colonoscopy and virology screen were done only when clinically indicated. Autoimmune screen was only performed if other autoimmune conditions were present. The most recent evidence from KDIGO 2021 is still to perform a secondary work up regardless the antibody test result.

3)Were histopathologic features of kidney biopsy considered (e.g. immunofluorescence staining pattern, location(s) of immune deposits, presence of other concomitant renal lesions)?

Our findings were not specifically correlated with histological features on biopsy other than the diagnosis of MN itself and we did not stain for PLA2R1 antigen on the renal tissue

4) Any data on concordance rate of serum PLA2R-ab vs. tissue PLA2R staining on biopsy specimen? Any cases exhibiting "immune sink" phenomenon?

We did not stain for PLA2R1 antigen on the renal tissue to correlate that with serum antibody test.

5) Renal biopsies are obtained for both diagnostic and prognostic purposes. Thus, it would be worthwhile to investigate the performance of serum PLA2R-Ab while considering other parameters (e.g. %global and segmental glomerulosclerosis, % tubulointerstitial scarring, tubulointerstitial inflammation, Ehrenreich-Churg stage of MN)

This would be beyond the scope of our study. We agree that this might be of useful prognostic value, but not for diagnostic purposes. Our study was mainly focusing on the diagnostic accuracy of the antibody test.

---

## [Decision Letter · Decision Letter 1]

1 Jul 2022

PONE-D-21-33835R1Can Use of the serum anti-PLA2R antibody negate the need for a renal biopsy in Primary Membranous Nephropathy?

PLOS ONE

Dear Dr. Ragy,

Thank you for submitting your manuscript to PLOS ONE. After careful consideration, we feel that it has merit but does not fully meet PLOS ONE’s publication criteria as it currently stands. Therefore, we invite you to submit a revised version of the manuscript that addresses the points raised during the review process.

In the light of the Reviewer 2 comments we believe that at least  a subset of samples should be stained for PLA2R1 antigen on the renal tissue to correlate that with serum antibody test. A correlation with histopathologic features of kidney biopsy should be considered, as well.

We look forward to receiving your revised manuscript.

Kind regards,

Fabio Sallustio, PhD

Academic Editor

PLOS ONE

Reviewers' comments:

Reviewer's Responses to Questions

**Comments to the Author**

1. If the authors have adequately addressed your comments raised in a previous round of review and you feel that this manuscript is now acceptable for publication, you may indicate that here to bypass the “Comments to the Author” section, enter your conflict of interest statement in the “Confidential to Editor” section, and submit your "Accept" recommendation.

Reviewer #1: All comments have been addressed

Reviewer #2: (No Response)

2. Is the manuscript technically sound, and do the data support the conclusions?

Reviewer #1: (No Response)

Reviewer #2: No

3. Has the statistical analysis been performed appropriately and rigorously? 

Reviewer #1: (No Response)

Reviewer #2: I Don't Know

4. Have the authors made all data underlying the findings in their manuscript fully available?

Reviewer #1: (No Response)

Reviewer #2: Yes

5. Is the manuscript presented in an intelligible fashion and written in standard English?

Reviewer #1: (No Response)

Reviewer #2: Yes

6. Review Comments to the Author

Reviewer #1: (No Response)

Reviewer #2: I am disappointed to learn that pathologic parameters from the biopsy were not considered in your study to determine if serum antiPLA2R Ab test should replace kidney biopsy in diagnosis of MN. While PLA2R+ MN is most often a PMN, other investigators have reported cases of PLA2R+ MN associated with HBV infection (Am J Nephrol 2015;41(4-5):345-53.) and PLA2R+ MN associated with malignancy (Clin Kidney J 2015; 8: 433-9). Moreover, there is also report of low prevalence serum antiPLA2R ab in patients with membranous lupus nephritis (Lupus. 2019 Mar;28:396-4050). Thus, I am in disagreement with the author's conclusions that "diagnosis of primary membranous nephropathy can rely solely on a positive PLA2 R-Ab test without the need for renal biopsy"

7. PLOS authors have the option to publish the peer review history of their article (what does this mean?). If published, this will include your full peer review and any attached files.

Reviewer #1: **Yes: **Fabrizio D'Ascenzo

Reviewer #2: No

---

## [Author Response · Author response to Decision Letter 1]

8 Jan 2023

Dear PLOS, ONE reviewer

Many thanks for the time given to review our work and for providing us with your valuable feedback. 

The utility and validity of pathological parameters in kidney biopsy tissue is a salient research question, including PLA2R staining; others including IgG4 vs staining IgG1 staining (1), neutrophils in glomeruli (2). We would however wish to note that staining PLA2R1 for histopathological samples to correlate with serum antibody test is beyond the scope of our study, and we believe it will not necessarily answer the primary question on the diagnostic utility of serum PLA2R-Ab testing. We hypothesize that the serum PLA2R-Ab test can replace the gold standard renal biopsy (using conventional stains, with PLA2R staining being unavailable or yet to be standardised in many centres). The objective of our study was to compare the reference test (standard renal biopsy) without PLA2R stain vs the index test (serum PLA2R-Ab test that is readily available as a commercial test) to determine if serum PLA2R-Ab testing can replace the renal biopsy. 

Whilst we agree with reviewer #2 that in certain situations, the PLA2R-ab test can be falsely positive and that a tissue diagnosis may provide more insight, staining for PLA2R1 in addition to serum testing will only likely improve the sensitivity with diagnosis. This question was studied by other investigators in previous investigations. Larsen et al, 2013, examined PLA2R1 in renal tissues in 165 cases of MN, including 85 primary MN and 80 secondary MN, and found tissue staining to have a sensitivity of 75% and a specificity of 83% (3). Hill et al, found that combining both biopsy and serum testing only improved the diagnostic sensitivity of the test, but not the specificity. In the aforementioned study, the specificity of the PLA2R-Ab test alone was 100% and the sensitivity was 81% compared to 100% and 89.5% respectively for tissue staining. When they combined both tissue and serum testing, the sensitivity improved to 95.2%, but the specificity was already 100% without the need for a biopsy (5). Therefore, we do agree that a biopsy in situations where the PLA2R-Ab test was negative remains an indispensable test, but not when the anti-PLA2R antibody test in serum were to be positive. In our current study, only 2 subjects had a false positive result out of 93 subjects, with a specificity of 97.9%. Even the presence of tissue staining for PLA2R or IgG4 can only raise the possibility that such cases are pathogenically related to primary MN than secondary MN (3). Many other investigators noted the similar association, especially with Hepatitis B and Sarcoidosis. 

Debiec and Ronco et al, have shown that some patients may have positive serum PLA2R- Ab test and their antigen stain on biopsy is negative, which is another challenging question on why the biopsy stain would not correlate with serum testing (6). We would argue that the complexity of the immune system and antigenicity of the podocyte is not fully discovered, and you would agree with us that in those situations even if a biopsy was performed, it might not change the clinician’s decision to treat.

Also, one would appreciate that even the renal biopsy being the gold standard diagnostic test for many years, it will not always provide 100% specificity and in some situations, it might miss a true diagnosis due to various reasons. Therefore, tissue or serum positivity would only suggest PLA2R association, but not being a primary or secondary driven pathology (4). Differentiation between Primary and Secondary may still require standard evaluation with clinical history, examination and screening for secondary causes.

We have reviewed the papers you have cited for our attention, which state that HBV, SLE, and cancers can result in a falsely positive PLA2R-Ab test. This observation is true, but only reported in a small number of studies. Performing a renal biopsy for those small number of patients needs to be weighed against the other larger population who could potentially avoid renal biopsy with its downstream costs (e.g., relating to complications including additional inpatient stays in hospital, blood transfusions, interventions post-bleeding complications), and the societal cost of biopsy (e.g., healthcare cost, patients and carer’s taking time off work). Moreover, this directs us to highlight that all patients who present with nephrotic syndrome, will have had a routine screening for HBV and SLE at disease presentation, which may already underpin the diagnosis of those conditions even before the PLA2R-Ab test results are available.

Having both HBV serology and PLA2R-Ab test positive, one may consider either a diagnosis of Primary membranous nephropathy (PMN) complicated by HBV or secondary membranous nephropathy (SMN) due to HBV (7). In this scenario, a renal biopsy can only aid the diagnosis if PLA2R1, c1q, and IgG subclasses can be stained, which is not widely available in clinical practice. Hence, a routine renal biopsy without the use of those special stains might not add value to the serum PLA2R-Ab test. 

MN secondary to SLE has also been a challenging diagnosis even after the discovery of PLA2R1 antigen. Svobodovo et al, found that none of their 16 Czech patients with a biopsy confirmed diagnosis of SMN due to SLE had a PLA2R1 positive stain on the renal tissue (8). These findings strongly suggest that lupus MN is not always related to PLA2R-Abs. In that situation, although this might not be PLA2R1-driven pathogenesis, we would urge a renal biopsy given the negative serum PLA2R-Ab testing. In other scenarios where the serum PLA2R-Ab test was positive in a patient with SLE, we would highly recommend a secondary workup including ANA, anti-DNAs, and complement levels, which is clearly stated in our conclusion in manuscript line number 219 and 220 ‘further screening for secondary causes of MN should also be considered in specific cases’

We cannot ascertain that a renal biopsy would necessarily diagnose PLA2R-associated MN triggered by a secondary cause. Radice et al identified 7 cases of cancer-associated MN confirmed with biopsy. All had their serum PLA2R-Ab tested positive, yet none of those renal biopsies could help identify that malignancy can be the culprit in the MN diagnosis. (9).Lefaucheur et al used a cut-off of eight cells per glomerulus to distinguish malignancy-related MN cases from controls, they calculated the specificity as 75% and sensitivity as 92% with an area under the curve of 0.92(2). This would highlight that performing biopsy would not stand a better chance to diagnose malignancy compared to the serum PLA2R -Ab test sensitivity and specificity. To make the diagnosis of cancer-related MN even more challenging, in many situations MN diagnosis would precede the cancer diagnosis. Again, a biopsy in those scenarios would not add value to positive PLA2R-Ab serum testing. Nonetheless, cancer screening for the high-risk group would be urged as suggested in our conclusion.

We hope that through extensive literature review, we demonstrated that even if the serum PLA2R-Ab test was falsely positive in limited scenarios, a biopsy performed in clinical practice using the conventional stains might not add much value to the diagnosis. Also, we emphasize the importance of considering PMN with another independent disease like cancer, SLE, or HBV infection in situations where both the serum PLA2R-Ab and screening tests were positive.

References:

1. Huang, C. C., Lehman, A., Albawardi, A., Satoskar, A., Brodsky, S., Nadasdy, G., Hebert, L., Rovin, B., & Nadasdy, T. (2013). IgG subclass staining in renal biopsies with membranous glomerulonephritis indicates subclass switch during disease progression. Modern Pathology: An Official Journal of the United States and Canadian Academy of Pathology, Inc, 26(6), 799–805. https://doi.org/10.1038/modpathol.2012.237

2. Lefaucheur C, Stengel B, Nochy D, et al. Membranous nephropathy and cancer: Epidemiologic evidence and determinants of high-risk cancer association. Kidney Int. 2006;70(8):1510-1517. doi:10.1038/sj.ki.5001790

3. Larsen, C. P., Messias, N. C., Silva, F. G., Messias, E., & Walker, P. D. (2013). Determination of primary versus secondary membranous glomerulopathy utilizing phospholipase A2 receptor staining in renal biopsies. Modern Pathology: An Official Journal of the United States and Canadian Academy of Pathology, Inc, 26(5), 709–715. https://doi.org/10.1038/modpathol.2012.207

4. Disease, K. (2021). Kidney Disease: Improving Global Outcomes (KDIGO) Glomerular Diseases Work Group. KDIGO 2021 clinical practice guideline for the management of glomerular diseases. Kidney Int, 100(4S), S1–S276. https://doi.org/10.1016/j.kint.2021.05.021

5. Hill, P. A., McRae, J. L., & Dwyer, K. M. (2016). PLA2R and membranous nephropathy: A 3 year prospective Australian study: PLA2R and membranous nephropathy. Nephrology (Carlton, Vic.), 21(5), 397–403. https://doi.org/10.1111/nep.12624

6. Debiec, H., & Ronco, P. (2011). PLA2R autoantibodies and PLA2R glomerular deposits in membranous nephropathy. N Engl J Med, 364, 689–690)

7. Wang, R., Wu, Y., Zheng, B., Zhang, X., An, D., Guo, N., Wang, J., Guo, Y., & Tang, L. (2021). Clinicopathological characteristics and prognosis of hepatitis B associated membranous nephropathy and idiopathic membranous nephropathy complicated with hepatitis B virus infection. Scientific Reports, 11(1), 18407. https://doi.org/10.1038/s41598-021-98010-y

8. Svobodova, B., Honsova, E., Ronco, P., Tesar, V., & Debiec, H. (2013). Kidney biopsy is a sensitive tool for retrospective diagnosis of PLA2R-related membranous nephropathy. Nephrology, Dialysis, Transplantation: Official Publication of the European Dialysis and Transplant Association - European Renal Association, 28(7), 1839–1844. https://doi.org/10.1093/ndt/gfs439

9. Radice, A., Pieruzzi, F., Trezzi, B., Ghiggeri, G., Napodano, P., D’Amico, M., Stellato, T., Brugnano, R., Ravera, F., Rolla, D., Pesce, G., Giovenzana, M. E., Londrino, F., Cantaluppi, V., Pregnolato, F., Volpi, A., Rombolà, G., Moroni, G., Ortisi, G., & Sinico, R. A. (2018). Diagnostic specificity of autoantibodies to M-type phospholipase A2 receptor (PLA2R) in differentiating idiopathic membranous nephropathy (IMN) from secondary forms and other glomerular diseases. Journal of Nephrology, 31(2), 271–278. https://doi.org/10.1007/s40620-017-0451-5

---

## [Editor Report · Decision Letter 2]

27 Jan 2023

PONE-D-21-33835R2Can Use of the serum anti-PLA2R antibody negate the need for a renal biopsy in Primary Membranous Nephropathy?PLOS ONE

Dear Dr. Ragy,

Thank you for submitting your manuscript to PLOS ONE. After careful consideration, we feel that it has merit but it need minor revisions in the Abstract. Therefore, we invite you to submit a revised version of the manuscript that addresses the points raised during the review process.

The Abstract conclusions are overstated and should be in line with that reported at the end of the paper. Please, revise the Abstract conclusions highlighting that antibody test has the potential to allow for the diagnosis and treatment in up to 75% of PMN cases without the need for a renal biopsy but that patients with negative PLA2R-Ab tests will still require a biopsy to confirm their diagnosis.

We look forward to receiving your revised manuscript.

Kind regards,

Fabio Sallustio, PhD

Academic Editor

PLOS ONE

Journal Requirements:

Additional Editor Comments:

The Abstract conclusions are overstated and should be in line with that reported at the end of the paper. Please, revise the Abstract conclusions highlighting that antibody test has the potential to allow for the diagnosis and treatment in up to 75% of PMN cases without the need for a renal biopsy but that patients with negative PLA2R-Ab tests will still require a biopsy to confirm their diagnosis.
---

## [Author Response · Author response to Decision Letter 2]

29 Jan 2023

Dear reviewer

Thank you for considering our work . We have reviewed your comments, and agree with your suggestions about changing the abstract conclusion. You will find the new version dated 29/1/23 amended accordingly.

Many thanks for your help

Omar Ragy

Consultant Nephrologist

Manchester Institute of Nephrology and Transplantation

---

## [Editor Report · Decision Letter 3]

31 Jan 2023

Can Use of the serum anti-PLA2R antibody negate the need for a renal biopsy in Primary Membranous Nephropathy?

PONE-D-21-33835R3

Dear Dr. Ragy,

We’re pleased to inform you that your manuscript has been judged scientifically suitable for publication and will be formally accepted for publication once it meets all outstanding technical requirements.

Kind regards,

Fabio Sallustio, PhD

Academic Editor

PLOS ONE
---

## [Editor Report · Acceptance letter]

14 Feb 2023

PONE-D-21-33835R3 

Can Use of the serum anti-PLA2R antibody negate the need for a renal biopsy in Primary Membranous Nephropathy? 

Dear Dr. Ragy:

I'm pleased to inform you that your manuscript has been deemed suitable for publication in PLOS ONE. Congratulations! Your manuscript is now with our production department. 

Kind regards, 

on behalf of

Dr. Fabio Sallustio 

Academic Editor

PLOS ONE